Original research

# How local partnerships to improve urgent and emergency care have impacted delayed transfers of care from hospitals in England: an analysis based on a synthetic control estimation method

Gintare Malisauskaite ,[1] Karen Jones,[1] Stephen Allan,[1] Daniel Roland,[1] Yvonne Birks,[2] Kate Baxter,[2] Kate Gridley[2]

¹Personal Social Services Research Unit, University of Kent, Canterbury, UK
²Social Policy Research Unit, University of York, York, UK

**Correspondence to**
Dr Gintare Malisauskaite;
g.malisauskaite@kent.ac.uk

## ABSTRACT

**Objectives** Patients should be discharged from hospital when they are medically fit. However, discharges are often delayed for non-medical reasons including access to social care. One aim of local health and social care partnerships to improve urgent and emergency care in England (known as urgent and emergency care (UEC) vanguards) was to improve integration of health and social care, which could lead to fewer delays. Consequently, we aimed to assess the impact of UEC vanguards on delayed discharges from hospital (delayed transfers of care (DTOC)) in England.

**Design** Using a synthetic control estimation method 29 local authorities (LAs) that were UEC vanguards partners were averaged into a single 'treated' unit and compared with a unit created using data from LAs that were not UEC vanguards partners to estimate the impact of UEC vanguards on DTOC. Sensitivity analysis included fixed effects panel regressions and various placebo tests.

**Setting** 150 LAs in England (excluding city of London and Isles of Scilly); 29 LAs were partners in UEC vanguards between August 2015 and March 2018.

**Primary outcome measure** Quarterly data on days of DTOC at LA level for the period 2010–2017.

**Results** Synthetic control estimation showed a large difference in DTOC days between UEC vanguards partner LAs compared with those that were not, with on average 23.7% lower DTOC per quarter (491 DTOC days per quarter). Fixed effect panel regressions found DTOC rates lower by 43.1% (99% CI 13.8% to 72.4%) in UEC partner LAs after the start of the vanguards programme. We found no indication of UEC partner LAs having lower DTOC rates prior to initiation of vanguards.

**Conclusions** The evidence indicates a sizeable statistically significant impact of UEC vanguards on DTOC; however, more research is required to explain the underlying reasons for this relationship.

## BACKGROUND

Delayed transfers of care (DTOC) is a term used to describe situations where patients are medically fit to be discharged from hospital to home or further care settings but the process is

---

### Strengths and limitations of this study

► The long span of delayed transfers of care (DTOC) data used from all English local authorities (LA) created a comprehensive picture of how urgent and emergency care (UEC) vanguards affected DTOC rates.

► The synthetic control method allowed for a comparison between UEC and non-UEC LAs to be drawn allowing observation of how UEC partner LAs would have been likely to perform in DTOC days in the absence of UEC vanguards.

► The data did not permit the identification and estimation of the impact of different initiatives associated to the UEC vanguards or account for differences in breadth and reach of UEC vanguards within the partner LAs.

► The design did not permit a clear explanation of the mechanisms by which UEC vanguards influenced reduction in DTOC days.

---

delayed.[1 2] DTOC has attracted increased attention from policy makers alongside health and social care professionals in England[3] due to increased rates in recent years; there is an estimated £820 million annual cost for the population aged 65+ years (2015–2016 estimate).[4] DTOC is also associated with decreased subsequent participation in activities of daily living, frailty, ageing, high comorbidity, cognitive impairment and dependency.[5–11]

Attempts to address the costs associated with DTOC have inspired a number of innovative policy approaches to integrating social and healthcare, including the Better Care Fund,[12] Integrated Care Pioneers[13] and more recently the New Models of Care – Vanguards.[14] Vanguards set out to help improve integration of services with five different approaches[14]:

► *Acute care collaboration* – linking local hospitals to improve clinical and financial viability.
► *Urgent and emergency care* – improving coordination of services and reducing pressure on accident and emergency (A&E) departments.
► *Enhanced health in care homes* – improving and integrating health, care and rehabilitation services for older people in care homes.
► *Multispecialty community providers* – moving specialist care into the community from hospitals.
► *Integrated primary and acute care systems* – joining up general practitioners (GP), hospitals, community and mental health services.

This paper examines the relationship between urgent and emergency care (UEC) vanguards and DTOC rates specifically, since discharge planning from acute care in hospitals was identified as one of the challenges by the UEC vanguards.[15] Eight UEC vanguards were announced to take effect in July–August 2015 with a planned end date of March 2018; however, a substantial number of vanguards managed to provide enhanced services beyond that date.[16] Models of care adopted by UEC vanguards largely aimed to better integrate the different ways that urgent and emergency care could be accessed. To facilitate service planning and design and help patients access care via the most effective and efficient routes, UEC vanguards were encouraged to use a tool called a channel shift modelling tool.[17] The tool was aimed at facilitating further integration and cooperation between health, social and community care services and included planning for discharge from hospital from the point of admission.[14 15] Thus, DTOC was likely to be directly affected by improved discharge planning and communication between health and social care providers. In addition, DTOC rates may be a useful proxy for certain aspects of integration, and it has been previously used as a criterion for evaluating the success of health and social care integration policy initiatives.[18]

To date, research related to DTOC is somewhat scarce. Most research considers challenges related to the discharge of older people: appropriate future support, suitable discharge destinations and how policies are put into practice.[19–22] Waiting for posthospital care packages accounts for a large proportion of reasons for delays,[23] and lack of social care supply is considered a major part of the explanation behind DTOC.[24 25] However, while delays for which social care is responsible have increased substantially, so have delays for which the National Health Service (NHS) is deemed responsible.[26 27] Indeed, the NHS is continuously considered responsible for the majority of delays, with internal hospital issues with planning, documentation and transport being cited as explaining increases in DTOC,[28] alongside concerns over the integration of services more generally.[29 30] To contribute to this literature and as part of a wider study examining the role of social care in DTOC,[31] we examined the link between UEC vanguards and DTOC rates.

## METHODS
### Data and setting
Data were collected for 150 English local authorities (LAs) for the time period between 2010 quarter 4 and 2017 quarter 4 (150 LAs, 29 quarters, 4350 observations). City of London and the Isles of Scilly were excluded due to stark differences in size in comparison with other LAs. The analysis was carried out at quarterly LA level; the time frame chosen based on the availability of DTOC data at the time of analysis. The eight UEC vanguards took effect in August 2015 (calendar year quarter 3) and included 29 LAs as partners.[14] The start date of UEC vanguards was nominal, but without more precise information on exact timings of when the programmes took effect in different locations or when it was likely to expect any impact on outcomes, we used this nominal start date as the start of the 'treatment' in our analyses.

### Statistical analysis
The primary chosen method of analysis was synthetic control estimations.[32 33] This method creates a control unit that matches the main characteristics of the treated unit so that it has a similar outcome trajectory prior to the treatment. This then allows observation of how the 'untreated' control unit would have performed in the time following the start of the treatment, providing a comparison with the treated unit. In this case, the 'treatment' was participation in UEC vanguard programme, and the 'treated unit' refers to derived averages of the outcome measure and control variables,[34] of all 29 LA partners of UEC vanguards partner sites, that is, the 29 LAs at each quarter were used in the sample to create a single treated unit over time. The control unit was created using the remaining LAs in England (ie, non-UEC vanguard partners) by estimating different weights, chosen automatically by the synthetic control estimation algorithm, for LAs to account for changes in confounding variables over time as well as across LAs. Furthermore, we controlled for the outcome variable at each quarter prior to the treatment to achieve close tracking between the two units over the period prior to the treatment. This allowed for a credible prediction of the counterfactual, that is, what would have happened in the absence of the vanguards programme in the UEC partner LAs. DTOC trends not being parallel prior to vanguards makes synthetic control estimation a preferred method for evaluation since it does not require parallel trends assumption to hold.

The outcome measure used for synthetic control estimations was the number of DTOC days (including all delayed discharge days). The analysis controlled for the following factors associated with LA-level DTOC: (1) demographics and level of need (carer's allowance, disability living allowance, total population in LA and percentage of population above 65 years old), (2) LA structure (type, size in square metres, percentage of people living in rural areas, the number of clinical commissioning groups each LA is in partnership with and care home bed supply) and (e) economic variables (jobseeker's allowance, pension

credit, house prices and percentage of single occupancy older people home ownership). A similar set of control variables were selected by Roland et al,[35] based on findings from the literature,[5–9 19–23] concerning confounders of delayed discharges from hospitals and data availability. Further details of the data are available in online supplemental appendix table A1.

## Sensitivity analysis

As a sensitivity check, we also estimated two-way fixed effects panel regressions (difference-in-difference approach). Any statistically significant results would further strengthen the argument of the existence of the relationship between UEC vanguards and DTOC. This provided an average estimate of the effect size of the vanguards programme on DTOC. The model specification used was:

$$DTOC_{it} = \alpha_i + V_{it}\beta_1 + V_{it}T_t\beta_2 + C_{it}\beta_3 + u_{it} \qquad (1)$$

where the dependent variable $DTOC_{it}$ is expressed as the natural logarithm of the number of DTOC days to mitigate the potential effect of large outliers and data skewness in regression results. Subscript $i$ indicates an LA, $t$ indicates time (in quarters) and $\alpha$ is a LA dependent intercept. Vanguard partners were identified using the dummy variable $V_{it}$ (1=after programme start, 2015 quarter 3, 0=before or not vanguard partner), and $\beta_1$ is the coefficient of interest for the analysis, showing the average effect of being a partner in the UEC vanguards on DTOC. We also included interactions between participation in UEC vanguards and time quarters $V_{it}T_t\beta_2$, with $\beta_2$ being a vector of coefficients associated with them. Finally, $C_{it}$ is a vector of the same control variables used in the synthetic control estimation and quarter dummies for each quarter in the sample, with $\beta_3$ being a vector of coefficients associated with them, representing the effects on DTOC rate.

The equation (1) was estimated using fixed effects panel regressions.[36] A Hausman test found that a fixed effects model was preferred over a random effects model (Hausman test statistic of 46.30, significant at 1%). Some variables were dropped during the estimation process due to the invariant nature of some data (eg, LA type and size, etc). All regressions used cluster-robust SEs centred on LAs to account for potential heteroscedasticity and unobservable characteristics that could make LA level clusters more similar.

Finally, given the potential for selection bias into UEC vanguards, as robustness checks we also conducted placebo tests to check if there was any evidence of significant differences in DTOC rates between UEC and non-UEC LAs by: (A) assuming that the UEC programme was in existence across the whole period of analysis and (B) using data from prior to the start of the programme. These robustness checks were estimated using ordinary least squares and random effects given UEC vanguard partners were assumed fixed over time.

| Table 1 | Descriptive statistics | | |
|---|---|---|---|
| Variable | Mean | SD | Min/Max |
| Dependent variables: | | | |
| DTOC days | 2755.394 | 3124.008 | 0/26733 |
| DTOC (log) | 7.438 | 0.986 | 0/10.194 |
| Explanatory variables: | | | |
| UEC vanguard | 0.067 | 0.249 | 0/1 |
| JSA ratio | 0.026 | 0.016 | 0.001/0.09 |
| PC ratio (65+) | 0.235 | 0.096 | 0.062/0.691 |
| CA ratio | 0.011 | 0.004 | 0.004/0.027 |
| DLA ratio (65+) | 0.087 | 0.040 | 0.025/0.257 |
| Care home beds (log) | 7.553 | 0.825 | 5.451/9.461 |
| Population (log) | 12.598 | 0.606 | 10.519/14.257 |
| Population 65+ ratio | 0.166 | 0.043 | 0.06/0.286 |
| Rurality (%) | 17.507 | 24.468 | 0/100 |
| No. of CCGs to LA | 5.660 | 3.480 | 1/21 |
| House prices (£, log) | 12.343 | 0.519 | 11.443/14.62 |
| Owning single home ratio (65+) | 0.075 | 0.022 | 0.013/0.126 |
| Owning single home outright ratio (65+) | 0.069 | 0.021 | 0.011/0.115 |
| Area (m²) | 86839.430 | 150126.3 | 1213/803761 |
| CCG dummy | 0.655 | 0.475 | 0/1 |
| LA type: | | | |
| Metropolitan | 0.240 | 0.427 | 0/1 |
| London | 0.213 | 0.410 | 0/1 |
| County | 0.180 | 0.384 | 0/1 |
| No. of obs. | 4350 | | |

Further information on data sources and derivation of variables available in online supplemental appendix table A1.
CA, carer's allowance; CCGs, Clinical Commissioning Groups; DLA, disability living allowance; DTOC, delayed transfers of care; JSA, jobseeker's allowance; LA, local authority; PC, pension credit; UEC, urgent and emergency care.

## Patient and public involvement statement

Both public and practice stakeholders were involved in the project as part of the steering committee. They helped to shape initial study design and participated in the discussion of initial and final findings alongside dissemination. Further stakeholder engagement through dissemination at practice facing events looking at findings further shaped the presentation of early findings.

## RESULTS
### Descriptive statistics

Table 1 presents descriptive statistics from the last quarter of 2010 to the last quarter of 2017. The average number of DTOC days was 2755 per English LA per quarter. A percentage of 6.7% of the sample observations were UEC vanguard partner sites. Table 2 provides more detail on outcome and control variables based on participation in UEC vanguard programme presenting mean values

**Table 2** Means of outcome and control variables by UEC vanguard participation, before and after the start of the programme

| | Non-UEC | | UEC | |
| --- | --- | --- | --- | --- |
| | **Before** | **After** | **Before** | **After** |
| DTOC days | 2510.955 | 3690.847 | 2130.523 | 2233.072 |
| DTOC (log) | 7.344334 | 7.758407 | 7.255489 | 7.297711 |
| JSA ratio | 0.031272 | 0.012509 | 0.038485 | 0.016309 |
| PC ratio (65+) | 0.254087 | 0.190102 | 0.263469 | 0.19152 |
| CA ratio | 0.010333 | 0.012434 | 0.011437 | 0.014433 |
| DLA ratio (65+) | 0.087383 | 0.080628 | 0.097996 | 0.090038 |
| Care home beds (log) | 7.521928 | 7.535296 | 7.666286 | 7.666745 |
| Population (log) | 12.60945 | 12.63799 | 12.5038 | 12.52737 |
| Population 65+ ratio | 0.161205 | 0.168712 | 0.174282 | 0.183642 |
| House prices (£, log) | 12.34778 | 12.52908 | 12.03327 | 12.17675 |
| Rurality (%) | 16.31793 | | 22.46746 | |
| No. of CCGs to LA | 5.752066 | | 5.275862 | |
| Owning single home ratio (65+) | 0.074886 | | 0.077415 | |
| Owning single home outright ratio (65+) | 0.068836 | | 0.071335 | |
| Area (m$^2$) | 83 452.44 | | 100 971.3 | |
| No of obs. | 2420 | 1210 | 551 | 290 |

Before: 2010 q4 to 2015 q2, 19 quarters; after: 2015 q3 to 2017 q4, 10 quarters.
Further information on data sources and derivation of variables available in online supplemental appendix table A1.
CA, carer's allowance; CCGs, Clinical Commissioning Groups; DLA, disability living allowance; DTOC, delayed transfers of care; JSA, jobseeker's allowance; LA, local authority; PC, pension credit; UEC, urgent and emergency care.

before and after the start of UEC vanguards. Before and after means for control variables appear similar for both UEC and non-UEC partner sites, the main noteworthy difference being the sizeable increase in DTOC days after the start of the vanguards in non-UEC LAs.

Figure 1 compares average number of DTOC days (ratio to LA population) per quarter for UEC partner LAs compared with those non-UEC LAs. The increase in DTOC days after the start of the UEC programme,

obvious for the non-UEC LAs, was hardly visible for LAs in UEC vanguards. However, since the trends prior to the vanguards seem non-parallel, robustness checks were employed to check for any indication that selection into the UEC vanguards programme was based on DTOC rates.

### Main results

Figure 2 depicts the predictions of synthetic control estimations. The figure shows that from the beginning of the UEC vanguards programme, partner sites had consistently lower than average DTOC rates than would have been

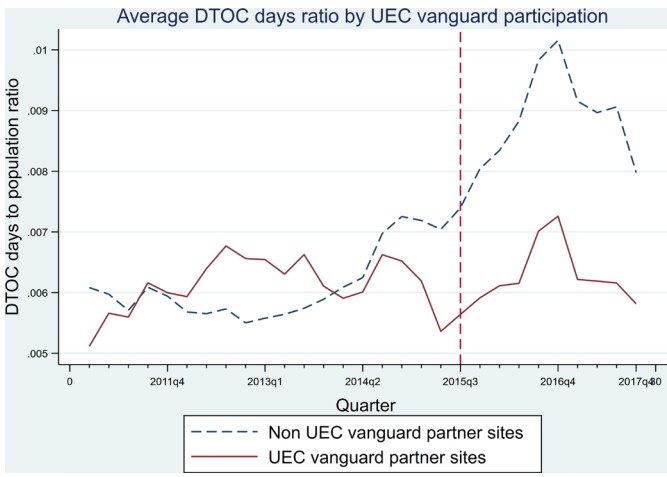

**Figure 1** Average DTOC days over time. Note: UEC vanguard partner sites include 29 LAs.[14] DTOC, delayed transfers of care; LAs, local authorities; UEC, urgent and emergency care.

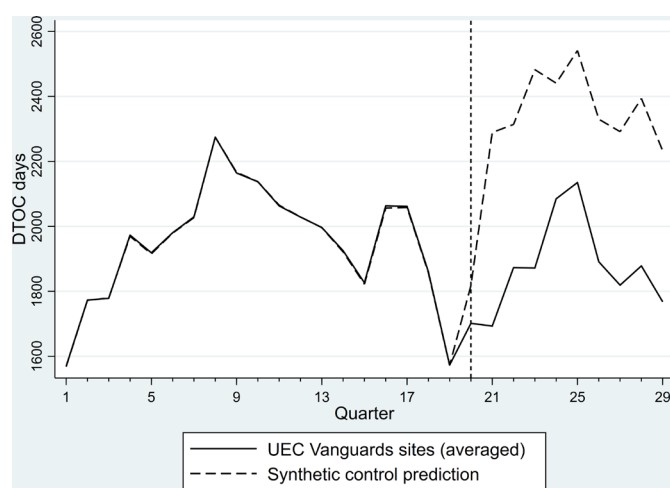

**Figure 2** Synthetic control estimation.

predicted without the vanguards. This difference was sizeable, with on average 491 fewer DTOC days per quarter per LA, or around 23.7% less (estimation outcome matrix, showing actual average days of DTOC for UEC partner LAs and predicted synthetic control unit days of DTOC, is provided in online supplemental appendix table A2). The synthetic control tracking the data prior to UEC programme indicates a good fit for the model (the predictor balances between averaged treated unit and synthetic control unit are available in online supplemental appendix table A3), and standardised p values indicate the quarterly effects are significant at 1% level (synthetic control postestimation results in online supplemental appendix table A4).[32 37] The sizeable dip in DTOC days just prior to the start of UEC vanguards of, on average, 300 delayed days less per UEC vanguard participating LA compared with the previous quarter, suggests that some form of preparation with regard to integration and DTOC could have taken place in the participating sites.

### Sensitivity analysis

The first column of table 3 reports the full results from the fixed effects panel regression estimation and shows UEC vanguards are associated with 43.1% lower DTOC rates at 1% significance level (99% CI 13.8% to 72.4%), and there was no indication of any trend differences between UEC partner and non-partner LAs after the start of the programme. It was a preferred estimation for finding the coefficient size associated with UEC vanguards impact, since when controlling for LA level characteristics Hausman test revealed systematic differences between fixed effects and random effects models and fixed effects model has an advantage of controlling for any other unobservable LA level characteristics that are fixed in time.

One of the possible interpretations for the observed results may be that sites participating in UEC vanguards could have been selected based on better DTOC rates. There was no publicly available information on specific selection criteria, but testing for statistical differences between DTOC rates was carried out by running the model using two modifications: (A) without identifying the start of the UEC vanguard, checking for an overall difference between UEC participating sites and others over the entire period of time of data; and (B) using only data prior to 2015, the start of the initiative.

The findings (table 3, columns A and B, respectively) showed there was no statistically significant relationship between UEC vanguards and DTOC in these specifications. Overall marginal effects of UEC vanguards for these specifications were also consistently statistically insignificant. This suggests that the difference in DTOC rates between UEC and non-UEC partner LAs only becomes significant after the start of UEC programme.

### DISCUSSION

#### Main findings

Our findings suggest that LAs that were part of UEC vanguards had significantly lower average DTOC than non-UEC sites after the start of the programme. Overall DTOC rates rose substantially in the second quarter of 2015 until the end of 2016, potentially explained by severe cuts to the funding of social care[38]; however, they rose significantly less, if at all, in UEC vanguard sites. We found no evidence suggesting that UEC vanguard sites had lower DTOC rates prior to becoming vanguards but are unable to rule out the possibility that they might have been in a better position to reduce DTOC due to other as yet unidentified factors, for example, other healthcare programmes already in place in the UEC partner LAs. It is difficult to pinpoint the reasons why UEC vanguards influenced DTOC rates. One potential explanation from policies and initiatives associated with being in a vanguard is the Channel Shift modelling tool kit, which supports integration of services, communication and cooperation between hospital and community based services.[18] Channel Shift interventions include planning discharge from time of admission, discharging for further assessment ('discharge to assess') and rapid response services, yet we could not formally test if this indeed is the main reason for the effect found.

There is limited previous research that has looked at the relationship between new models of care – vanguards and DTOC rates. One study reported that vanguards are associated with a small reduction in hospital admissions.[39] This highlights important implications for the National Health Service and social care, alongside individuals discharged from hospital. The move towards greater integration of services is not a new idea, and the importance of local collaboration has been consistently stressed.[40] However, it is difficult to isolate the impact of the UEC vanguards from other government integration policies, together with the austerity climate facing the care sector. This is particularly important with regard to potential changes the COVID-19 response has led to, including hospital discharge service, which is expected to influence the further development of discharge to assess and integration of health and social care services.[41] Although new care models were discontinued well before the start of the global pandemic, assumptions were made that practices adopted during the time of new care models activity period would continue to be used.[16] This makes it reasonable to expect this programme could have made a positive contribution towards freeing hospital beds for patients with COVID-19; however, further research would be necessary to establish the size and significance of any impact. Alongside our findings, this indicates that the vanguards programme should be of interest to policy makers in terms of lessons learnt for dealing with discharges during and following the COVID-19 pandemic.

#### Limitations

The synthetic control estimation suggests an element of causality between UEC participation and DTOC rates. However, a number of limitations need to be acknowledged. First, we could not account for selection criteria into the vanguards, different initiatives within UEC

**Table 3** Regression results with DTOC (log) as dependent variable

| Variable | FE | RE (A) | RE (B) | OLS (A) | OLS (B) |
|---|---|---|---|---|---|
| UEC vanguard | −0.431*** (0.112) | −0.0852 (0.150) | −0.054 (0.149) | −0.101 (0.149) | −0.053 (0.147) |
| JSA ratio | −2.949 (5.746) | −4.046 (4.572) | −0.393 (4.330) | 1.229 (4.802) | 10.734* (6.065) |
| PC ratio (65+) | 3.559 (2.429) | 2.783** (1.414) | 1.478 (1.391) | 1.229 (0.848) | 0.832 (0.979) |
| CA ratio | 31.956 (51.607) | −4.278 (25.682) | −4.517 (29.333) | −41.223** (18.047) | −60.315** (26.653) |
| DLA ratio (65+) | −1.668 (7.182) | −4.941** (2.244) | −2.106 (2.498) | −0.063 (1.616) | 1.209 (1.756) |
| Care home beds (log) | −0.539 (0.353) | −0.401* (0.213) | −0.289 (0.229) | −0.187 (0.161) | −0.264 (0.197) |
| Population (log) | 1.950 (2.111) | 1.658*** (0.266) | 1.629*** (0.269) | 1.405*** (0.198) | 1.540*** (0.239) |
| Population 65+ ratio | 16.315 (10.441) | 12.130** (5.602) | 11.702** (5.950) | 2.336 (3.246) | 2.641 (4.652) |
| House prices (£, log) | 0.028 (0.263) | 0.107 (0.146) | 0.161 (0.136) | 0.070 (0.133) | 0.195 (0.159) |
| CCG dummy | 0.317 (0.419) | 0.363* (0.216) | −0.034 (0.123) | 0.715** (0.159) | 0.556*** (0.149) |
| Rurality (%) | − | −0.009 (0.006) | −0.010 (0.008) | −0.005 (0.005) | −0.006 (0.008) |
| No. of CCGs to LA | − | −0.024* (0.014) | −0.033** (0.015) | −0.031*** (0.012) | −0.033** (0.014) |
| Home single ownership ratio (65+) | − | −105.012* (56.140) | −160.919*** (61.456) | −74.033* (40.551) | −126.771** (50.798) |
| Home single ownership outright ratio (65+) | − | 104.434* (56.140) | 160.952*** (60.303) | 81.398* (41.406) | 137.968*** (50.871) |
| Area (m²) | − | −3.25e-08 (4.76e-07) | 7.95e-09 (7.34e-07) | 1.86e-07 (5.48e-07) | 2.30e-07 (8.33e-07) |
| LA type: | | | | | |
| Metropolitan | − | −0.308** (0.152) | −0.431*** (0.140) | −0.202* (0.103) | −0.345*** (0.123) |
| London | − | −0.502*** (0.159) | −0.586*** (0.162) | −0.602*** (0.120) | −0.665*** (0.142) |
| County | − | 0.149 (0.203) | 0.120 (0.211) | 0.230 (0.183) | 0.211 (0.203) |
| Cons. | −17.234 (28.685) | −10.257*** (2.839) | −13.516*** (2.592) | −10.062*** (2.181) | −12.676*** (2.629) |
| Interactions between time quarters and UEC vanguard | Yes | Yes | Yes | Yes | Yes |
| Time quarter dummies | Yes | Yes | Yes | Yes | Yes |
| No. of obs. | 4350 | 4350 | 2550 | 4350 | 2550 |
| No. of groups | 150 | 150 | | 150 | |

Robust clustered SEs in square brackets underneath the coefficients. Variables omitted during the FE estimation process due to fixed effects nature of the model: LAs' rurality (%), no. of CCGs to LA, owning single home ratio (65+), owning single home outright ratio (65+), area (m²), LA type; more details of variable construction available in the online supplemental appendix.

Robustness checks in RE and OLS columns: (A) UEC vanguard partner LAs are identified as being active with a binary variable across the whole period of analysis; (B) UEC vanguard partner LAs are identified with a binary variable only using sample prior to 2015. FE results not available for these specifications due to treatment being fixed across time in the estimations.
***p<1%, **p<5%, *p<10%.
CA, carer's allowance; CCG, Clinical Commissioning Group; DLA, disability living allowance; JSA, jobseeker's allowance; LAs, local authorities; OLS, ordinary least squares; PC, pension credit; RE, random effect; UEC, urgent and emergency care.

vanguards and the impact of other active health policies potentially influencing the results during the vanguards' activity period. Such information could further inform analysis and provide further context for this finding. Further work on specific policies used within UEC vanguard sites, including qualitative analysis and more detailed examination of specific UEC vanguards' mechanisms of action could help untangle potential reasons for the association between UEC vanguards and DTOC.

Second, this analysis does not account for different timeframes in which UEC vanguards took effect in different locations, differences within the eight UEC vanguards

or possible different levels of exposure to UEC vanguard influence in each LA. We anticipate there was some variation in associations between different UEC vanguards and outcomes. However, this approach should be sufficient to show the average effect of the UEC vanguards programme.

Third, we did not look into readmission rates for UEC vanguard partner sites. Evidence suggests that there is an association between high bed occupancy and readmission rates into hospitals in England, likely due to increased rates of discharge when bed occupancy is high.[42] There remains the possibility that hospitals in UEC partner LAs may have had higher readmission rates because of higher bed occupancy and/or improved rates of discharge. If so, the positive benefit of the UEC vanguard programme would be mitigated somewhat. However, the UEC vanguard programme directly addressed one of the likely causes of readmissions from high bed occupancy, that is, inadequate discharge planning. The identified limitations could be further routes of enquiry, even if achieving precision in quantifying healthcare programmes is unlikely.

A potential solution to refining the analysis would be to use individual-level data to conduct a similar analysis. This could allow more precise estimation of the effect size as case-specific confounders could be taken into account, which is not possible at an aggregated level.

Overall, the estimation methods adopted revealed a strong association between the UEC vanguard and a reduction of DTOC, which advocates the success of this integration programme and should encourage further research to reveal which specific aspects of this initiative were particularly beneficial, which could guide further policy decisions.

**Contributors** GM conceived the statistical methodology, performed the statistical analysis, drafted the manuscript and is responsible for the overall content as the guarantor; KJ contributed to the conception, study design and the final writing of this paper; SA contributed to the study design, statistical methodology and the final writing of this paper; DR contributed to the statistical methodology and the final writing of this paper; YB, KB and KG critically revised the draft and contributed to the final writing of this paper.

**Funding** This study was funded by the National Institute for Health Research School for Social Care Research, grant number C088/CM/UKJB-P116.

**Disclaimer** The views expressed in this independent research publication are those of the authors and not necessarily those of the NIHR SSCR, the National Institute for Health Research or the Department of Health and Social Care.

**Competing interests** None declared.

**Patient consent for publication** Not applicable.

**Ethics approval** This study does not involve human participants.

**Provenance and peer review** Not commissioned; externally peer reviewed.

**Data availability statement** Data are available on reasonable request.

for any error and/or omissions arising from translation and adaptation or otherwise.

**ORCID iD**
Gintare Malisauskaite http://orcid.org/0000-0002-0797-5949

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
