## [Reviewer comments · BMJ Open]

ARTICLE DETAILS

TITLE (PROVISIONAL)	How local partnerships to improve urgent and emergency care have impacted delayed transfers of care from hospitals in England: an analysis based on a synthetic control estimation method
AUTHORS	Malisauskaite, Gintare; Jones, Karen; Allan, Stephen; Roland, Daniel; Birks, Yvonne; Baxter, Kate; Gridley, Kate

VERSION 1 – REVIEW

REVIEWER	Okuno, Hideo Department of Critical Care Medicine, Osaka City General Hospital
REVIEW RETURNED	07-Sep-2021

GENERAL COMMENTS	The manuscript entitled “Impact of Urgent and Emergency Care Vanguard on Delayed Transfers of Care in England” is a well written and important contribution to the literature. The authors approach the relationship between DTOC and UEC Vanguard programme using synthetic control estimations. They mention some limitations and check the robustness. There are a few comments that should be addressed prior to publication including: 1. The authors should mention the role of UEC Vanguard more clearly in the background section. We can understand the relationship between the UEC Vanguard and DTOC in this manuscript. As the authors mention in the discussion section, however, it is difficult to consider why the UEC Vanguard influence DTOC. I think additional description of UEC Vanguard would be more helpful to think about the causality, especially for the Non-UK readers.2. The authors should add some information to Table 1 and Table 2. Rural or hub means the percentage of residents in the rural area? Owning house ratio (65+) in Table 1 means 7.5% of 65+ people have the home ownership? I am afraid it is too low (https://www.brookings.edu/essay/uk-rental-housing-markets/). Area (m2) in Table2 means the average area of the local authorities in Non-UEC and UEC?
--

REVIEWER	Ling, Tom RAND corporation
REVIEW RETURNED	21-Sep-2021

GENERAL COMMENTS	This article reports on an important set of results in response to a question that remains highly relevant. The paper appropriately emphasizes that further work is needed to to understand the
---

	mechanisms that might explain these results. However, the apparent scale of the results demonstrates the value for both researchers and policy makers of further work in this area. The limitations are clearly laid out. There is one well known issue that might helpfully be flagged in the conclusion - that reductions in DTOC may also be associated with higher re-admission rates. In the light of what we now know about the response to COVID-19 it might also be of interest to readers to reflect a little further on how these results are still relevant for policy makers and researchers..
--	---

REVIEWER	Wang, Shaolin The University of Manchester
REVIEW RETURNED	29-Sep-2021

GENERAL COMMENTS	This paper aims at evaluating impact of Urgent and Emergency Care Vanguards on an interesting new outcome - delayed transfers of care. The authors use synthetic control method to create a control unit that matches the main characteristic of the treated unit so that it has similar outcome trajectory prior to the treatment and found UEC LAs on average had 507(21%) fewer DTOC days than non-UED LAs. They also use fixed effects panel regressions and found DTOC rates were 40.5% lower in UEC LAs. The paper touches an important topic, as most evaluations of New Models of Care has focused on hospital admissions, few on discharge planning. However, the methodology and empirical analysis need to be refined for the results to be more compelling. Below I provide some specific comments that may be helpful to the authors when refining the paper. 1. Start of the 'treatment' The authors used the nominal start date as the start of the 'treatment' in the analyses, while from Figure 1 we could see that hospitals might have started to make changes and reduce the DTOC rates since 2014q3. I would suggest to treat 2014q3-2015q3 as the "rollout" period of the Vanguards programme and match the control group based on the outcome trajectory of the treatment group before this time period. 2. Statistical analysis Considering "synthetic control estimation does not produce a single estimate of the effect size of the Vanguards programme on DTOC", the authors estimate fixed effects panel regressions to "provide a single estimate of the treatment size for reference purposes." Did the authors estimate the fixed effects panel regression on the original sample? If so, is the fixed effects panel regression essentially a Difference-in-Difference (DiD) analysis? DiD could be a suitable method in this case because if we treat 2014q3-2015q3 as the "rollout" period as suggested above, we may not have to reject the parallel trends prior to the start of Vanguards programme. In addition, Figure 1 suggests that both groups saw increase in DTOC days within the first five quarters after programme start and then decrease in the following three quarters. I would suggest extending the fixed effects model by including quarter or year dummy variables and their interaction with the dummy variable for
--

	Vanguard partners instead of one single post-programme. The will allow the two groups following different time trends after programme start and could test whether the treatment size varies over time. Last but not least, synthetic control estimation produces estimates of the effect size for each quarter, and the authors average treatment effects over the period. Therefore I am not convinced that the authors need to run fixed effects regression due to the requirement to produce a single estimate of the treatment size. There could have additional justifications. The problem of the synthetic control method is that it applies a different inference framework and does not construct the traditional standard errors. For comparative inference the authors will need to generate standard errors using placebo tests. For reference, the Placebo approach was proposed by Abadie et al in the paper “Synthetic Control Methods for Comparative Case Studies: Estimating the Effect of California’s Tobacco Control Program” published in Journal of the American Statistical Association in 2010. One minor comment is to tick each quarter and label consistently on the X-axis of Figure 1 and 2.
--	--

VERSION 1 – AUTHOR RESPONSE

Reviewer: 1

Dr. Hideo Okuno, Department of Critical Care Medicine, Osaka City General Hospital Comments to the Author:

The manuscript entitled “Impact of Urgent and Emergency Care Vanguard on Delayed Transfers of Care in England“ is a well written and important contribution to the literature. The authors approach the relationship between DTOC and UEC Vanguard programme using synthetic control estimations. They mention some limitations and check the robustness. There are a few comments that should be addressed prior to publication including:

Thank you very much for your kind and useful comments helping us revise and improve the manuscript.

1. The authors should mention the role of UEC Vanguard more clearly in the background section. We can understand the relationship between the UEC Vanguard and DTOC in this manuscript. As the authors mention in the discussion section, however, it is difficult to consider why the UEC Vanguard influence DTOC. I think additional description of UEC Vanguard would be more helpful to think about the causality, especially for the Non-UK readers.

We are grateful for this insight and amended the Background section of the article to help explain the anticipated link between UEC Vanguard and DTOC better (pgs. 4-5)

2. The authors should add some information to Table 1 and Table 2. Rural or hub means the percentage of residents in the rural area? Owning house ratio (65+) in Table 1 means 7.5% of 65+ people have the home ownership? I am afraid it is too low (<https://www.brookings.edu/essay/uk-rental-housing-markets/>). Area (m2) in Table 2 means the average area of the local authorities in Non-UEC and UEC?

Thank you for pointing out a few things we can clarify. We now included a note under Table1 and Table2:

‘Further information on data sources and derivation of variables available in Appendix TableA1.’

Rural or hub was renamed to ‘Rurality (%)’ in Tables 1 and 2, to indicate the % of population in rural areas, this is a fixed indicator in time with data from 2011.

Both of the variables regarding home ownership are for single occupancy house ownership for 65+ as a ratio to all households in a Local Authority: owning with mortgage and owning outright, with data coming from year 2011, ONS. The data is correct, it was a mistake on our part how it was explained, thank you for spotting this! Clarified in tables, text and Appendix.

In Table 2 Area (m²) indicates separate means for UEC and non-UEC Local Authorities in the sample (left side non-UEC, right side UEC); Table 2 summarizes the control variables before and after the start of UEC Vanguard by participation in the programme, but some variables are fixed in time in our sample, so their means did not vary before and after the UEC Vanguard.

Reviewer: 2

Dr. Tom Ling, RAND corporation

Comments to the Author:

This article reports on an important set of results in response to a question that remains highly relevant. The paper appropriately emphasizes that further work is needed to understand the mechanisms that might explain these results. However, the apparent scale of the results demonstrates the value for both researchers and policy makers of further work in this area. We are very thankful for your generous review and for flagging a couple more things we could improve.

The limitations are clearly laid out.

There is one well known issue that might helpfully be flagged in the conclusion - that reductions in DTOC may also be associated with higher re-admission rates.

Thank you for pointing this out to include in our limitations section. On pg. 14 we now include:

'Thirdly, we did not look into re-admission rates for UEC Vanguard partner sites. Evidence suggests that there is an association between high bed occupancy and re-admission rates into hospitals in England, likely due to increased rates of discharge when bed occupancy is high,(42). There remains the possibility that hospitals in UEC partner LAs may have had higher readmission rates because of higher bed occupancy and/or improved rates of discharge. If so, the positive benefit of the UEC Vanguard programme would be mitigated somewhat. However, the UEC Vanguard programme directly addressed one of the likely causes of readmissions from high bed occupancy, i.e. inadequate discharge planning.'

In the light of what we now know about the response to COVID-19 it might also be of interest to readers to reflect a little further on how these results are still relevant for policy makers and researchers.

It is a good idea to reflect on this further and thank you for this suggestion. We added a consideration of what this impact may be and why it might interest researchers and policy makers (pg. 12):

'This makes it reasonable to expect this programme could have made a positive contribution towards freeing hospital beds for Covid-19 patients, however further research would be necessary to establish the size and significance of any impact.'

Reviewer: 3

Dr. Shaolin Wang, The University of Manchester Comments to the Author:

This paper aims at evaluating impact of Urgent and Emergency Care Vanguard on an interesting new outcome - delayed transfers of care. The authors use synthetic control method to create a control unit that matches the main characteristic of the treated unit so that it has similar outcome trajectory prior to the treatment and found UEC LAs on average had 507(21%) fewer DTOC days than non-UEC LAs. They also use fixed effects panel regressions and found DTOC rates were 40.5% lower in UEC LAs. The paper touches an important topic, as most evaluations of New Models of Care has focused on hospital admissions, few on discharge planning. However, the methodology and empirical analysis need to be refined for the results to be more compelling.

Below I provide some specific comments that may be helpful to the authors when refining the paper. We truly appreciate your insightful and detailed suggestions, please see our discussion of how we addressed each point separately below.

1. Start of the 'treatment'

The authors used the nominal start date as the start of the 'treatment' in the analyses, while from Figure 1 we could see that hospitals might have started to make changes and reduce the DTOC rates since 2014q3. I would suggest to treat 2014q3-2015q3 as the "rollout" period of the Vanguard

programme and match the control group based on the outcome trajectory of the treatment group before this time period.

We are grateful for the suggestion, it is an interesting avenue to explore. We run a synth model with suggested 'rollout' period specification for comparison reasons, presented below. This estimation suggests even a larger difference between the treated unit (UEC Vanguard sites) and synthetic control unit, with synthetic prediction peaking above 3500 days compared to under 2600 in our original estimation. However, we feel it is better to stay with the formal start date indicated in UEC Vanguard's programme in our main analysis, since it is difficult to make a clear justification for assuming an earlier start of the programme, or how much earlier it should start (a guess that would mainly be based on the behaviour of the data). Furthermore, by working with the official starting date of UEC Vanguard's for our analyses, we anticipate it would only reduce the estimate of the average effect of the UEC Vanguard's programme on DTOC (supported by results below), and we believe a more conservative estimate is more appropriate considering other discussed limitations of our study.

2. Statistical analysis

Considering “synthetic control estimation does not produce a single estimate of the effect size of the Vanguard's programme on DTOC”, the authors estimate fixed effects panel regressions to “provide a single estimate of the treatment size for reference purposes.”

Did the authors estimate the fixed effects panel regression on the original sample? If so, is the fixed effects panel regression essentially a Difference-in-Difference (DiD) analysis? DiD could be a suitable method in this case because if we treat 2014q3-2015q3 as the “rollout” period as suggested above, we may not have to reject the parallel trends prior to the start of Vanguard's programme.

Many thanks for your considerations and clarifying questions that help us explain our method better. Fixed effects regressions were run on the sample identified in the manuscript, the full sample between last quarter of 2010 and last quarter of 2017, for all 150 Local Authorities (4,350 observations), with no missing values. This indeed would make it a DID analysis. The issue with parallel trends was one of the reasons why we did not use FE regressions for main analysis and rather used it as a robustness check. It is true, that if we extended the period of time when we assume UEC Vanguard's are active to include another 4 quarters prior to 2015 quarter3, parallel trends issue would be potentially forgone. However, it is difficult to find a justification for why we should assume that this UEC Vanguard's programme has been active for a year before its official start date and not any other time period. This is why we consider synthetic control estimations to be our main analysis since it does not require for parallel trends assumption to hold, and we use FE regressions as supplementary to the main analysis.

In addition, Figure 1 suggests that both groups saw increase in DTOC days within the first five quarters after programme start and then decrease in the following three quarters. I would suggest extending the fixed effects model by including quarter or year dummy variables and their interaction with the dummy variable for Vanguard partners instead of one single post-programme. This will allow the two groups following different time trends after programme start and could test whether the treatment size varies over time.

We appreciate the suggestion. We re-run analysis using the suggested approach, also including a single estimate UEC variable besides the interactions of quarter dummies with UEC Vanguard indicator. We believe it is important to allow for the single estimate of the average impact of the UEC programme in addition to interaction terms. Interaction terms would show any programme related changes over time and their statistical significance. This has also been adopted for RE and OLS placebo tests regressions.

We present the full results of this re-estimation below. We find the UEC dummy is highly statistically significant and higher than estimated previously (-0.431), but we find no significance for the interaction terms (if estimated without the single estimate of the average effect, then all interaction terms are statistically significant).

Fixed-effects (within)							
regression	Number of obs	=		4,350			
Group variable: lacode	Number of groups	=		150			
R-sq:	Obs per group:						
within = 0.2302	min	=		29			
between = 0.6617	avg	=		29			
overall = 0.5692	max	=		29			
	F(46,149)	=		10.37			
corr(u_i, Xb) = -0.4295	Prob > F	=		0			
(Std. Err. adjusted for 150 clusters in lacode)							
				Robust			
DTOC(log)	Coef.	Std. Err.	t	P>t	[95% Conf.	Interval]	
UEC Vanguard	-0.4310903	0.112182	-3.84	0.000	-0.65276	0.2094166	
UEC Vanguard #quarter							
1 1		0 (empty)					
1 2		0 (empty)					
1 3		0 (empty)					
1 4		0 (empty)					
1 5		0 (empty)					
1 6		0 (empty)					
1 7		0 (empty)					
1 8		0 (empty)					
1 9		0 (empty)					
1 10		0 (empty)					
1 11		0 (empty)					
1 12		0 (empty)					
1 13		0 (empty)					
1 14		0 (empty)					
1 15		0 (empty)					
1 16		0 (empty)					
1 17		0 (empty)					
1 18		0 (empty)					
1 19		0 (empty)					
1 20	0.0638621	0.090967	0.7	0.484	-0.11589	0.2436135	
1 21	0.0614573	0.086126	0.71	0.477	-0.10873	0.2316429	
1 22	0.0728101	0.09206	0.79	0.43	-0.1091	0.2547215	
1 23	-0.0093152	0.096096	-0.1	0.923	-0.1992	0.1805726	
1 24	-0.0019042	0.086237	-0.02	0.982	-0.17231	0.168501	
1 25	0.0120544	0.076633	0.16	0.875	-0.13937	0.163483	
1 26	-0.0104569	0.056595	-0.18	0.854	-0.12229	0.1013751	

1 27	-0.024386	0.065762	-0.37	0.711	-0.15433	0.1055615
1 28	-0.0539734	0.04704	-1.15	0.253	-0.14692	0.0389778
1 29	0	(omitted)				
JSA ratio	-2.94891	5.746421	-0.51	0.609	-14.3039	8.406093
PC ratio (65+)	3.558504	2.428962	1.47	0.145	-1.24116	8.358166
CA ratio	31.95598	51.60663	0.62	0.537	-70.0194	133.9313
DLA ratio (65+)	-1.668025	7.181788	-0.23	0.817	-15.8593	12.52328
Care home beds (log)	-0.5393244	0.352692	-1.53	0.128	-1.23625	0.1575998
Population (log)	1.949947	2.110984	0.92	0.357	-2.22139	6.121279
Population 65+ ratio	16.31475	10.44061	1.56	0.12	-4.31602	36.94553
Rurality (%)	0	(omitted)				
No CCGs to LA	0	(omitted)				
House prices (£, log)	0.0279838	0.263128	0.11	0.915	-0.49196	0.5479286
Owning single home r. (65+)	0	(omitted)				
Owning single home outr. r. (65+)	0	(omitted)				
Area(m ²)	0	(omitted)				
LA type:						
Metropolitan	0	(omitted)				
London	0	(omitted)				
County	0	(omitted)				
CCG dummy quarter	0.3165302	0.418944	0.76	0.451	-0.51131	1.14437
2	0.0112157	0.046374	0.24	0.809	-0.08042	0.1028508
3	-0.0500636	0.066634	-0.75	0.454	-0.18173	0.0816062
4	0.0400965	0.076296	0.53	0.6	-0.11066	0.1908577
5	0.039386	0.081096	0.49	0.628	-0.12086	0.1996328
6	-0.0073044	0.133538	-0.05	0.956	-0.27118	0.2565687
7	0.0132897	0.150934	0.09	0.93	-0.28496	0.3115381
8	0.0553172	0.162471	0.34	0.734	-0.26573	0.376362
9	0.0293205	0.172116	0.17	0.865	-0.31078	0.3694237
10	0.0084806	0.219587	0.04	0.969	-0.42543	0.4423866
11	-0.2848717	0.225506	-1.26	0.208	-0.73048	0.1607317
12	-0.2643327	0.209585	-1.26	0.209	-0.67848	0.14981
13	-0.248121	0.214641	-1.16	0.25	-0.67226	0.1760133
14	-0.2648699	0.183407	-1.44	0.151	-0.62728	0.0975444
15	-0.2165815	0.167549	-1.29	0.198	-0.54766	0.1144984
16	-0.1226727	0.151868	-0.81	0.421	-0.42276	0.1774194
17	-0.1133819	0.155644	-0.73	0.467	-0.42094	0.194172
18	-0.1420919	0.12953	-1.1	0.274	-0.39805	0.1138616
19	-0.1834566	0.119022	-1.54	0.125	-0.41865	0.0517325
20	-0.068247	0.109146	-0.63	0.533	-0.28392	0.1474266
21	0.0202402	0.107563	0.19	0.851	-0.19231	0.2327868
22	0.0289009	0.088699	0.33	0.745	-0.14637	0.2041718
23	0.0964981	0.081202	1.19	0.237	-0.06396	0.2569534
24	0.1940783	0.075236	2.58	0.011	0.045412	0.3427451
25	0.2307303	0.061132	3.77	0	0.109932	0.3515285
26	0.1235063	0.043354	2.85	0.005	0.037839	0.2091734
27	0.0974558	0.038435	2.54	0.012	0.021508	0.1734037
28	0.13029	0.026049	5	0	0.078817	0.1817629
29	0	(omitted)				
_cons	-17.23421	28.68499	-0.6	0.549	-73.9161	39.44772
sigma_u	0.57068231					
sigma_e	0.40326899					
rho	0.66695784	(fraction of variance due to u_i)				

Last but not least, synthetic control estimation produces estimates of the effect size for each quarter, and the authors average treatment effects over the period. Therefore I am not convinced that the authors need to run fixed effects regression due to the requirement to produce a single estimate of

the treatment size. There could have additional justifications. The problem of the synthetic control method is that it applies a different inference framework and does not construct the traditional standard errors. For comparative inference the authors will need to generate standard errors using placebo tests. For reference, the Placebo approach was proposed by Abadie et al in the paper “Synthetic Control Methods for Comparative Case Studies: Estimating the Effect of California’s Tobacco Control Program” published in Journal of the American Statistical Association in 2010. Thank you for the suggested reference, consequently we ran placebo tests and report post-estimation results with standardized p-values for reference, these show the effect sizes for the post-treatment period are significant at 1% level. They are reported in TableA4 in the Appendix. You are right, using FE regressions was meant not only for producing a single estimate of the effect size but also for estimating the statistical significance of the result. It also works as a good robustness check, i.e. if using two different estimation techniques we get a sizeable statistically significant result in both cases, this only further shows the importance of the finding (clarified in the manuscript too).

One minor comment is to tick each quarter and label consistently on the X-axis of Figure 1 and 2. A much appreciated suggestion! Done as requested.

VERSION 2 – REVIEW

REVIEWER	Wang, Shaolin The University of Manchester
REVIEW RETURNED	17-Dec-2021

GENERAL COMMENTS	The paper has been well refined. The authors apply synthetic control method and generate standard errors using placebo tests. I like the idea of using fixed effects regression (DiD) and various placebo tests as sensitivity analysis. My only question is the about the key parameters of the DiD analysis. In the current DiD model, “Vanguard partners were identified using the dummy variable Vit (1 = after programme start, 2015 quarter 3, 0 = before or not Vanguard partner)”, and “interactions between participation in UEC Vanguards and time quarters” were also included. Vit indicates post-intervention period for Vanguard partners, so it is essentially an interaction between participation in Vanguards and post-intervention. I think Vit and time quarter dummies would be enough for the DiD model. There is no need to include an additional interaction variables $VitTt$. One minor comment is to omit “model also includes year indicators between 2010-2017” on page 21. I understand the authors have used quarter dummies instead.
---

VERSION 2 – AUTHOR RESPONSE

Reviewer: 3

Dr. Shaolin Wang, The University of Manchester Comments to the Author:

The paper has been well refined. The authors apply synthetic control method and generate standard errors using placebo tests.

Thank you for your positive feedback and an attentive detailed consideration of our results.

I like the idea of using fixed effects regression (DiD) and various placebo tests as sensitivity analysis. My only question is the about the key parameters of the DiD analysis.

In the current DiD model, “Vanguard partners were identified using the dummy variable Vit (1 = after programme start, 2015 quarter 3, 0 = before or not Vanguard partner)”, and “interactions between participation in UEC Vanguards and time quarters” were also included. Vit indicates post-intervention period for Vanguard partners, so it is essentially an interaction between participation in Vanguards and post-intervention. I think Vit and time quarter dummies would be enough for the DiD model. There is no need to include an additional interaction variables $VitTt$.

Thank you for the comment. Yes, it is true, Vit represents interaction term and was our original estimation strategy. However, after including the additional interaction terms with quarters, we believe there is no necessity to remove them since i) they introduce additional information about if and how the effect of intervention varied over time, ii) do not qualitatively change the overall result and the change in the main coefficient is minor, iii) it is used as sensitivity analysis so these results are not used separately as indicative or conclusive.

One minor comment is to omit “model also includes year indicators between 2010-2017” on page 21. I understand the authors have used quarter dummies instead.

Thank you so much for noticing this, done as requested.